# Sensitive detection of SARS-CoV-2 seroconversion by flow cytometry reveals the presence of nucleoprotein-reactive antibodies in unexposed individuals

Leire Egia-Mendikute [1,8], Alexandre Bosch [1,8], Endika Prieto-Fernández [1], So Young Lee[1], Borja Jiménez-Lasheras[1], Ana García del Río[1], Asier Antoñana-Vildosola[1], Chiara Bruzzone[2], Maider Bizkarguenaga[2], Nieves Embade[2], Rubén Gil-Redondo [2], María Luz Martínez-Chantar[3,4], Marcos López-Hoyos[5], Nicola G. A. Abrescia [4,6,7], José M. Mato [2,4], Óscar Millet[2] & Asís Palazón [1,7 ✉]

There is an ongoing need of developing sensitive and specific methods for the determination of SARS-CoV-2 seroconversion. For this purpose, we have developed a multiplexed flow cytometric bead array (C19BA) that allows the identification of IgG and IgM antibodies against three immunogenic proteins simultaneously: the spike receptor-binding domain (RBD), the spike protein subunit 1 (S1) and the nucleoprotein (N). Using different cohorts of samples collected before and after the pandemic, we show that this assay is more sensitive than ELISAs performed in our laboratory. The combination of three viral antigens allows for the interrogation of full seroconversion. Importantly, we have detected N-reactive antibodies in COVID-19-negative individuals. Here we present an immunoassay that can be easily implemented and has superior potential to detect low antibody titers compared to current gold standard serology methods.

[1] Cancer Immunology and Immunotherapy Lab, CIC bioGUNE, Basque Research and Technology Alliance (BRTA), Bizkaia, Spain. [2] Precision Medicine and Metabolism Lab, CIC bioGUNE, Basque Research and Technology Alliance (BRTA), Bizkaia, Spain. [3] Liver Disease Laboratory, CIC bioGUNE, Basque Research and Technology Alliance (BRTA), Bizkaia, Spain. [4] Centro de Investigación Biomédica en Red de Enfermedades Hepáticas y Digestivas (CIBERehd), Instituto de Salud Carlos III, Madrid, Spain. [5] Servicio Inmunología, Hospital Universitario Marqués de Valdecilla-IDIVAL, Facultad de Medicina, Universidad de Cantabria, Cantabria, Spain. [6] Structure and Cell Biology of Viruses Lab, CIC bioGUNE, Basque Research and Technology Alliance (BRTA), Bizkaia, Spain. [7] Ikerbasque, Basque Foundation for Science, Bizkaia, Spain. [8] These authors contributed equally: Leire Egia-Mendikute, Alexandre Bosch. ✉email: apalazon@cicbiogune.es

The severe acute respiratory syndrome coronavirus 2 (SARS-CoV-2) global spread has resulted in an ongoing pandemic[1]. To date, most immunoassays to determine seroconversion and measure antibody responses are based on enzyme-linked immunosorbent assay (ELISA), including automated chemiluminescent variants. Serological assays are important to detect previously infected individuals and perform epidemiological seroconversion studies[2–4]. Moreover, they have important implications in the development of antibody-based therapeutics (i.e. convalescent serum or monoclonal antibodies) and vaccines (i.e. selection of non-immunized individuals and follow-up). For these reasons, there is a need of developing fast and sensitive serology assays that can be deployed at a large scale[5].

SARS-CoV-2 contains several structural proteins, among them the Spike (S) and the Nucleoprotein (N) are the most immunogenic viral antigens and are used in serologic assays[6,7]. The S protein is comprised of two subunits: S1 and S2. S1 includes the receptor-binding domain (RBD) that binds to its cognate receptor angiotensin converting enzyme 2 (ACE2) expressed by host cells[8–10]. Its sequence is specific for SARS-CoV-2, often generating neutralizing antibodies in seropositive individuals[11]. Given their specificity, both RBD and S are considered ideal for serology assays[12–14], especially in the form of recombinant proteins produced in mammalian cell systems that reflect a physiological glycosylation pattern[15]. Humoral immunity against SARS-CoV-2 has been reported[16], including the presence of neutralizing antibodies in seropositive individuals[17]. Moreover, specific cellular responses have been described, including memory T cell formation against immunodominant peptides[18–20].

In general, ELISAs have an acceptable specificity and sensitivity profile for performing large epidemiological studies, but their sensitivity in the context of SARS-CoV-2 serology could be improved[21,22]. A key limitation of ELISA is the need of individual plates/wells for each antigen or antibody class to be tested. Moreover, the antigen is immobilized to the plate, which can hide epitopes or increase the background noise. For these reasons, ELISAs are not well suited to detect low antibody titres and often give undetermined values that are close to the cut-off, leading to difficult interpretation of results.

Cytometric bead arrays offer an alternative to perform serology. This technology allows for the rapid identification of multiple analytes simultaneously on a multiplexed manner, requiring less amounts of sample than traditional immunoassays[23]. Its reproducibility and sensitivity are well characterized, especially for measuring cytokines[24]. The readout is based on flow cytometry, open systems that are widely available in clinical and research settings.

In this study, we have developed a flow-cytometric bead array (C19BA) to assess seroconversion against SARS-CoV-2, leveraging the multiplex capability of this technology for the simultaneous interrogation of the presence of IgG and IgM antibodies against three viral antigens. This approach unravelled the presence of N-reactive antibodies in a cohort of samples collected before the pandemic, indicating that crossreactivity against this conserved viral protein exists.

## Results

### Development of a flow-cytometric bead array (C19BA) for the detection of SARS-CoV-2 seroconversion.
The presented flow cytometry assay consists in a multiplexed array containing microbeads with different intrinsic fluorescence intensities coated with viral antigens. The coupling was performed with microbeads functionalized with streptavidin and proteins tagged with a unique terminal biotin, which allows for the orientation of the antigen on the surface of the bead. The bead array (C19BA) is incubated with serum samples to allow the binding of anti-SARS-CoV-2 antibodies and then stained with anti-IgG and anti-IgM secondary antibodies labelled with different fluorochromes (Fig. 1a). In order to fully assess the specific seroconversion against SARS-CoV-2, we have chosen RBD, S1, and N as target antigens. The redundancy of RBD as a sequence included in S1 allows for the confirmation of intra-assay specificity on different microbeads simultaneously. The N protein was also included in the assay because of its immunogenicity. N is predicted to be less specific for SARS-CoV-2 based on the analysis of the sequence alignment with other coronavirus family members (Supplementary Fig. 1). We reasoned that fully seroconverted individuals would present antibodies against the three chosen antigens. C19BA includes uncoated negative control beads and positive control beads that are coated with human IgG and IgM (Fig. 1b). This setup differentiates each type of microbead as shown in the non-overlapping histograms in Fig. 1c. We first tested the ability of this assay to identify recombinant IgG antibodies against RBD and N. As can be seen in Fig. 1d, the microbead array clearly identified the binding of these antibodies. Importantly, the sensitivity of C19BA was superior to the ELISA presented here (Fig. 2) when their performance was compared in serial dilutions of commercial anti-RBD and anti-N IgG antibodies. C19BA presented a better broader dynamic range and identified low antibody concentrations that were not detected by ELISA.

### Determination of SARS-CoV-2 seroconversion on preCOVID-19 and acute COVID-19 cohorts by C19BA.
We then applied C19BA to interrogate serum samples from a cohort of 43 individuals who tested positive for SARS-CoV-2 infection by polymerase chain reaction (PCR). These samples were obtained at the time of hospital admission (acute COVID cohort). As a control, sera from 50 individuals collected before the pandemic (2018–2019) were analysed (preCOVID cohort). Both ELISA and C19BA were able to discriminate both cohorts based on the presence of IgG and IgM antibodies against RBD, S1, and N. At this early stage of infection, not all samples from the acute COVID cohort presented reactivity against viral antigens (Supplementary Fig. 2). Serial dilutions of 10 seropositive acute COVID and 10 preCOVID samples were performed to further compare the sensitivity of C19BA versus ELISA. Figure 3a shows the dilution curves corresponding to the presence of IgG antibodies by these two methods. C19BA was superior to the ELISA presented here separating both cohorts and identifying the presence of antibodies at lower concentrations. This was confirmed by plotting the area under the curve (AUC) and determining its statistical significance (Fig. 3b). The titres of IgM antibodies were lower than IgG as measured by both methods (Fig. 3c, d), in line with previous studies[16]. Importantly, our assay identified the presence of N-reactive IgG antibodies in six preCOVID samples ($n = 50$), although in general at lower titres than in the acute COVID cohort (Fig. 3a, b). Figure 4 shows representative dot plot profiles corresponding to preCOVID and COVID samples. While the reactivity against RBD and S1 was specific for COVID samples, crossreactivity against N was observed in some preCOVID individuals that contained IgG, but not IgM antibodies. No crossreactivity against SARS-CoV-2 RBD or S1 was observed on serum samples that tested seropositive against the spike of other common cold coronaviruses (Supplementary Fig. 3). In the

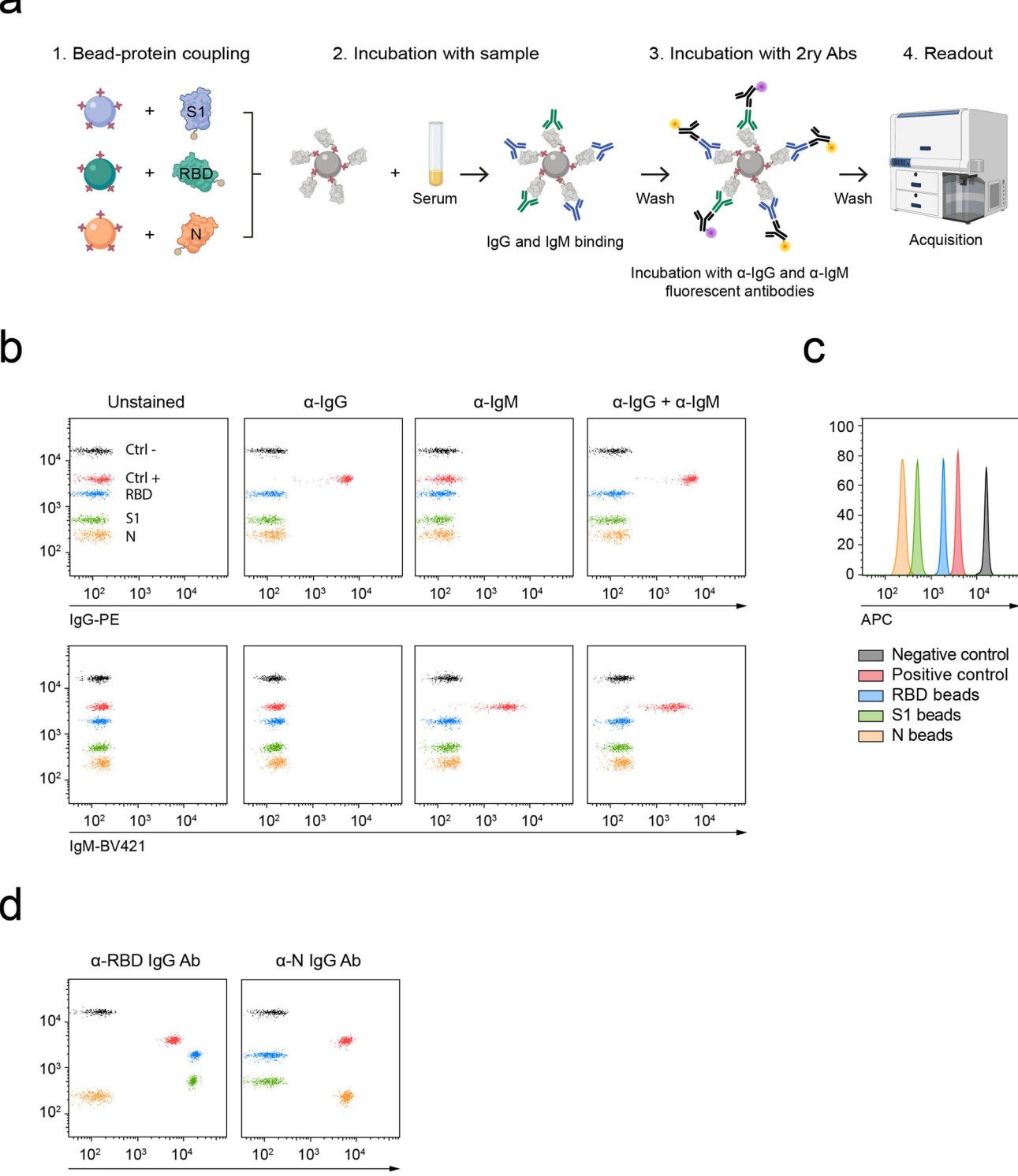

**Fig. 1 Workflow and performance of C19BA. a** Overview of the detection of SARS-CoV-2 seroconversion by flow-cytometric bead array. Streptavidin-coated microbeads with different fluorescence intensities are conjugated with recombinant biotinylated viral antigens (RBD, S1, and N) and mixed and incubated with pre-diluted serum samples together with control beads. After incubation, microbeads are washed and stained with anti-human IgG and IgM secondary antibodies, washed, and acquired on a flow cytometer for downstream analysis. Schematic created using BioRender.com. **b** Dot plots showing the staining patterns of positive (red) and negative (black) control beads with secondary anti-IgG-PE, anti-IgM-BV421, or both. The signal corresponding to IgG-PE (top) and IgM-BV421 (bottom) is shown for each column. **c** Histogram showing the distribution of the microbeads based on their intrinsic fluorescence on the APC channel. Each colour represents the coating for each microbead. **d** Representative dot plots showing the specificity of the staining pattern of recombinant anti-RBD IgG (left) or anti-N IgG (right) antibodies.

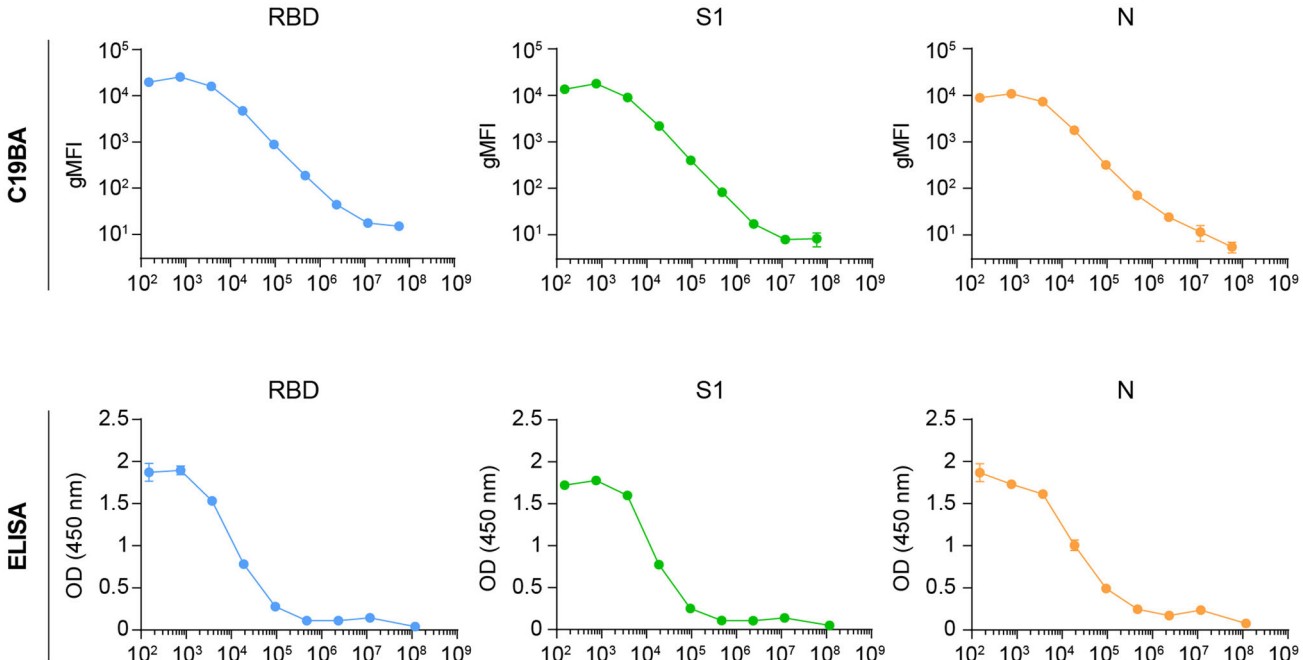

**Fig. 2 C19BA has superior sensitivity and greater dynamic range than ELISA.** Comparison of the titration of anti-RBD (blue and green) and anti-N (orange) IgG antibodies against recombinant RBD, S1, and N proteins by C19BA (top) and ELISA (bottom). The mean value of two replicates is shown; error bars represent SD.

acute COVID cohort, several samples that presented full seroconversion against RBD, S1, and N for both IgG and IgM were identified. A minority of COVID samples presented only N-reactive IgG and IgM antibodies, testing negative for RBD and S1 (Fig. 4).

**C19BA detects higher amounts of SARS-CoV-2 reactive antibodies in serum samples from severe COVID-19 patients compared to mild/moderate COVID-19 patients**. We then applied the C19BA assay on an additional set of serum samples obtained at time of hospital admission, corresponding to an independent cohort of patients classified by different clinical outcomes of the disease. These included mild/moderate ($n =$ 18) and severe ($n = 16$) cases that tested positive by PCR. Figure 5a shows that levels of anti-RBD and anti-S1 IgG antibodies measured by C19BA were significantly higher in severe COVID cases compared to mild/moderate cases and a control cohort comprising COVID-19-negative samples ($n = 18$). The same trend can be observed by ELISA but did not reach statistical significance. Figure 5b shows representative patterns of IgG seroconversion in COVID-19 seronegative samples, and mild/moderate and severe COVID-19 cases assayed by C19BA. Sensitivity and specificity values for each antigen analysed by C19BA were calculated by receiver operating characteristic (ROC) curve analysis of PCR positive and preCOVID samples (Supplementary Fig. 4).

**C19BA identifies the presence of IgG and IgM on convalescent individuals**. We next checked the ability of C19BA of measuring antibody levels on serum samples from convalescent individuals and compared those antibody levels to a seronegative control group. Figure 6a shows that C19BA detects the presence of anti-RBD, anti-S1, and anti-N IgG and IgM antibodies on serum samples from convalescent individuals. The seronegative control group lacked antibody reactivity against RBD and S1,

but 4 of 18 samples contained N-reactive IgG antibodies (Fig. 6a, b). This latest observation is in line with the previous findings on an independent preCOVID cohort (Supplementary Fig. 2 and Fig. 3a, b).

**Levels of anti-RBD and anti-S1 IgG antibodies measured by C19BA correlate with the neutralization capacity of serum samples**. We then studied the correlation between anti-SARS-CoV-2 antibody levels and the neutralization capacity of those antibodies. To this end, we tested seronegative (preCOVID, COVID negative) and seropositive (acute and convalescent) samples by C19BA and an ELISA-based inhibition assay that measures the binding of RBD to ACE2. Figure 7a shows that the neutralization capacity of seropositive samples was higher than seronegative samples. We explored correlations between the levels of anti-RBD, anti-S1, and anti-N IgG antibodies from seropositive individuals measured by C19BA and the inhibition of the interaction RBD/ACE2. RBD and S1 reactive antibody levels positively correlate with their inhibition capacity, while there is a lack of correlation in the case of N-reactive IgG levels (Fig. 7b).

**Discussion**
Sensitive characterization of humoral responses is critical to control the current pandemic[3], because it allows to perform accurate longitudinal serosurveys and epidemiological studies. Moreover, antibodies act as biomarkers for previous exposure and thus can guide vaccination strategies, including booster dosing. Serology assays need to be cost-effective, high-throughput, scalable, and easy to implement. In order to fulfil all these requirements, we have developed an assay that combines three viral antigens with superior sensitivity than the ELISAs presented here. The combination of antigens allows for the interrogation of full seroconversion, including the presence of antibodies against the partially redundant and specific

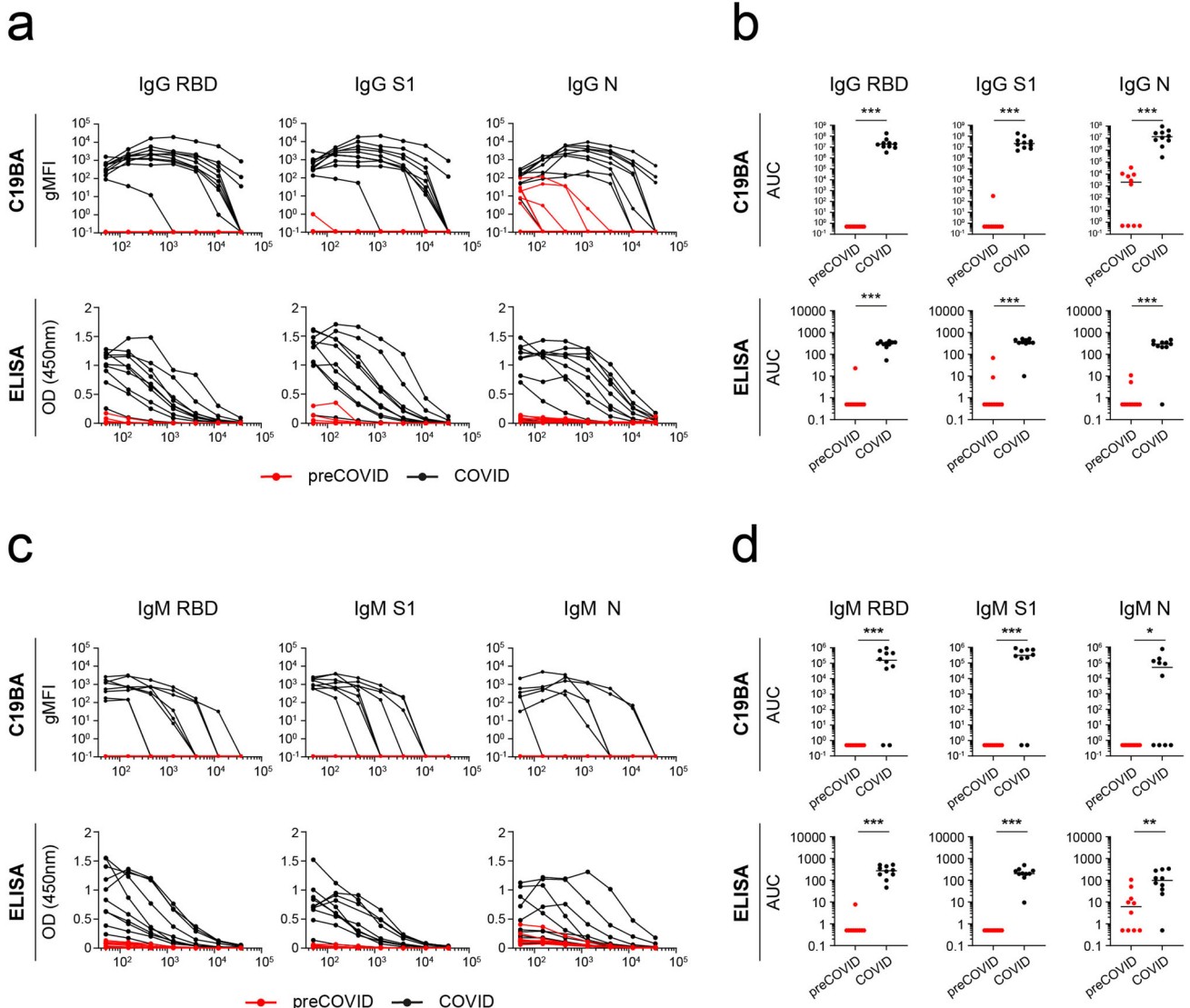

**Fig. 3 Identification of SARS-CoV-2 IgG and IgM seroconversion by C19BA. a** Titration curves of the reactivity of individual serum samples against RBD, S1, and N proteins measured by flow cytometry against IgG by C19BA (top) and ELISA (bottom) for each cohort: preCOVID (red, $n = 10$) and COVID (black, $n = 10$). COVID serum samples were obtained at time of hospital admission (PCR+). **b** AUC values for the experiment shown in **a**. **c** Titration curves of the reactivity of individual serum samples against RBD, S1, and N proteins against IgM by C19BA (top) and ELISA (bottom). **d** AUC values for the experiment shown in **c**. Statistical analyses were performed using an unpaired two-tailed Student's *t*-test. Asterisks represent *p* values (***$p < 0.001$, *$p < 0.05$, ns not significant). Horizontal lines represent median values.

RBD and S1 antigens, and the more conserved but strongly immunogenic N protein[25]. Together, these antigens offer a more specific and rapid platform than conventional assays that use only one viral antigen or require two-step sequential confirmation. In this context, RBD and S1 are specific for anti-SARS-CoV-2 antibodies and are not a source of crossreactivity for antibodies present in samples collected before the pandemic or sera containing antibodies against the spike of other coronaviruses.

Another key feature of robust serology assays is sensitivity, which is important to detect low antibody titres. ELISA assays often identify samples with low OD values that are close to the established cut-off, resulting in inconclusive or false-positive results. These require repetition of the assay and titration of samples. A previously developed ELISA includes a sequential

confirmatory assay with the Spike after positivity against RBD[12]. We demonstrate that C19BA presents a superior dynamic range than the ELISA presented here, and an improved limit of detection of low antibody titres. Indeed, C19BA is able to detect the presence of binding antibodies against three different viral antigens at dilutions that were beyond the limit of detection of ELISA.

Spike-reactive antibodies have neutralization capacity, even when binding non-RBD Spike epitopes[26]. On the other hand, N-reactive antibodies are not considered to confer protection and might even be predictive of poor patient outcome in some cases[27,28]. Levels of anti-RBD and anti-S1 IgG antibodies measured by C19BA on patient samples strongly correlate with the ability of these sera to neutralize the interaction between RBD and ACE2, as expected[29].

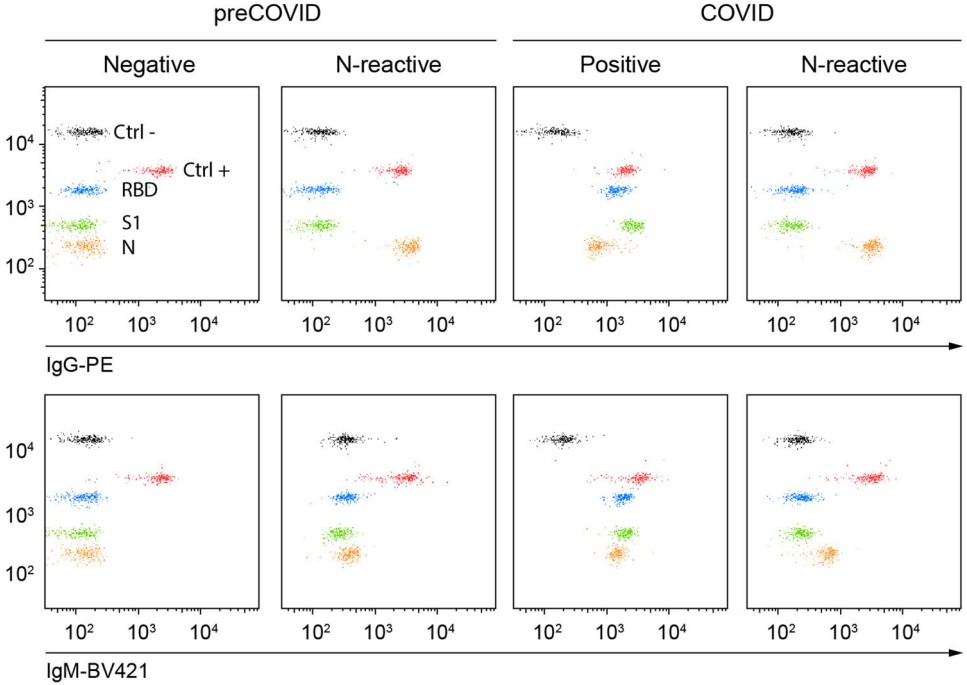

**Fig. 4 C19BA reveals different serology patterns in preCOVID and COVID samples collected at time of hospital admission (PCR+).** Dot plots showing IgG (top) and IgM (bottom) reactivity against RBD, S1, and N for representative preCOVID and COVID samples.

This superior sensitivity allowed for the identification of low titres of N-reactive antibodies in around 14% (10/68) of Covid-19 seronegative individuals; these antibodies were likely generated as a result of a previous common cold coronavirus infection. This fact raises important concerns about the specificity of the use of N protein for serological assays, given its high homology with N proteins of other coronaviruses. Indeed, several studies reported similar crossreactivity on serological testing based on N protein during the previous SARS-CoV outbreak in 2004, resulting in false positives[30,31].

Although most seropositive individuals in the COVID cohort presented antibodies reactive against RBD, S1, and N, a minority of samples only presented N-reactive antibodies. This suggests that these individuals either had antibodies from previous primary infections or mounted fast secondary antibody responses against N after mobilization of memory B cells generated as a result of a previous coronavirus infection. Recently, in a similar fashion, N-specific memory T cells have been also identified in COVID-19 unexposed individuals[18]. Together, these data provide evidence that cellular and humoral immune responses against SARS-CoV-2 exist as a result of crossreactivity against the N protein originated by previous coronavirus infections. The impact of these pre-existing T cell and antibody responses in the control and pathogenesis of COVID-19 requires further investigation.

In summary, we have developed a novel multiplexed method with higher sensitivity than traditional serology assays, using a triple combination of antigens that exploits the specificity of the Spike and RBD together with the less-specific N protein for the detection of antibodies against SARS-CoV-2.

## Methods

**Serum samples**. Serum samples corresponding to preCOVID and acute COVID individuals were provided by the Basque Biobank (www.biobancovasco.org) after approval from the corresponding ethics committee (CEIC-E 20-26, 1-2016). All participants in the study provided informed consent and were anonymized. The serum samples corresponding to the acute COVID cohort (43 patients presenting COVID-19 symptomatology and diagnosed by PCR) were obtained at the time of hospital admission. The preCOVID cohort (50 serum samples) was obtained during the yearly medical check-up of the working population of the Basque Country in 2018–2019 in collaboration with Osarten Kooperatiba Elkartea from Mondragon Corporation. Additional serum samples from negative controls ($n =$ 18) and independent COVID-19 cohorts confirmed by PCR were obtained after written informed consent and approval by the Cantabria Ethics Committee (CEIm Code: 2020.167). Serum samples from 34 patients with active infection were obtained at the time of hospital admission, and samples from 20 convalescent patients were obtained 1 month after recovery from COVID-19. Severity of the disease was defined as mild/moderate ($n = 18$) or severe ($n = 16$). Severity was classified based on admission to the intensive care unit and oxygen levels as defined previously[32].

**Microbead coating**. PMMA (polymethyl methacrylate) 8.2 μm microbeads coated with streptavidin were purchased from PolyAn (Cat#10652009). Each type of microbead presented a different fluorescence intensity (Red4 dye, excitation: 590–680 nm/emission: 660–780 nm). First, microbeads were washed with cold phosphate-buffered saline (PBS) pH 7.2 (Gibco Cat#14190-094) by centrifugation at 2000 r.p.m. for 5 min and resuspended in PBS. Then, biotinylated recombinant RBD, S1, and N (Acrobiosystems Cat#SPD-C82E9, Cat#S1N-C82E8, and Cat#-NUN-C81Q6, respectively) were added to the tubes (RBD at 11 μg/mL, S1 at 30 μg/mL, N at 19,5 μg/mL) and kept at 4 °C on a rotating head over tail for an hour, protected from light. Positive control beads were coated with biotinylated human IgG (Novus Biologicals Cat#NBP1-96855) and IgM (Novus Biologicals Cat#NBP1-96989) on the same microbead at 15 μg/mL each. Negative control beads were not coated with protein. After the coupling reaction, microbeads were washed three times with PBS. Then, D-biotin (2 μM) (Sigma-Aldrich Cat#8512090001) was added and incubated for 15 min at room temperature (RT) to inactivate residual streptavidin. After three additional washes, equal amounts of each microbead were combined in the same vial.

**C19BA assay**. Antigen-coupled microbeads were added to protein LoBind 1.5 mL Eppendorf tubes (Eppendorf Cat#525-0133) in a volume of 50 μL of PBS containing a total of 5000–6000 beads. After centrifugation (2000 r.p.m., 5 min), microbeads were resuspended with 100 μL of pre-diluted (PBS) serum samples or serially diluted commercial antibodies against RBD (GenScript Cat#A02038) or N (Acrobiosystems Cat#NUN-S41) starting from a 1 mg/mL concentration. Negative control samples were prepared with PBS. After a 30 min incubation (RT protected from light), samples were washed three times in PBS. Secondary antibodies were diluted in 100 μL of PBS containing 5% FBS: anti-human IgG-PE (1:50) (Clone

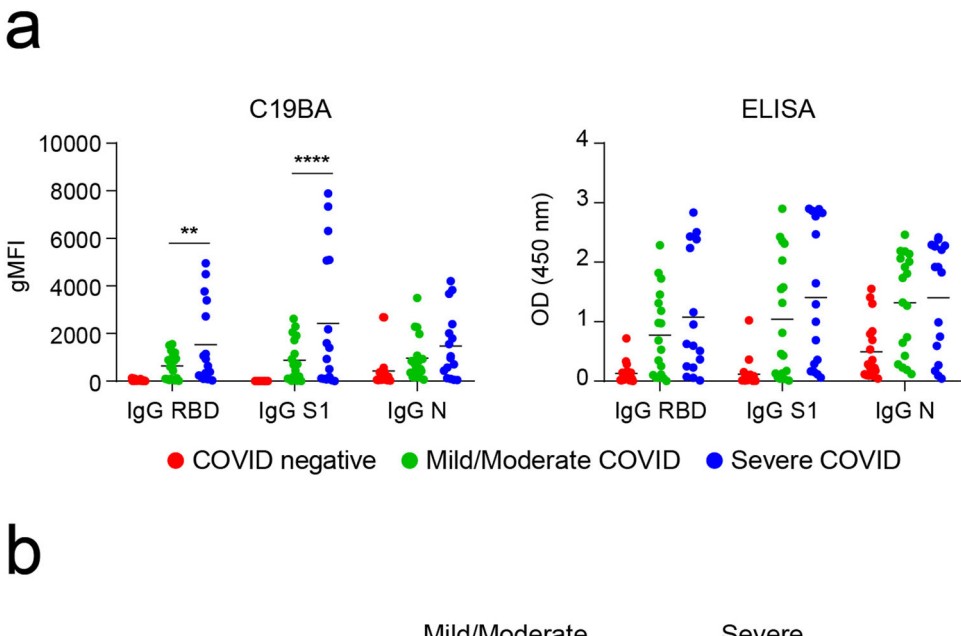

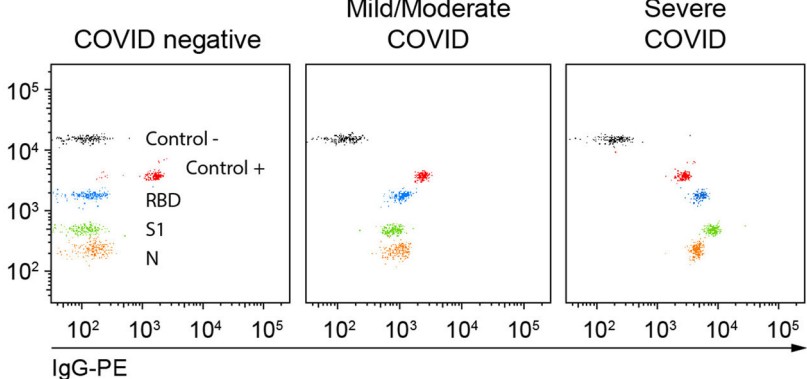

**Fig. 5 Severe Covid-19 patients present higher levels of anti-spike IgG antibodies compared to moderate Covid-19 patients. a** gMFI values obtained by C19BA (left) and OD values obtained by ELISA (right) corresponding to levels of IgG against RBD, S1, and N for the indicated cohorts. **b** Representative dot plots obtained by C19BA showing IgG levels for COVID negative ($n = 18$), mild/moderate COVID ($n = 18$), and severe COVID serums ($n = 16$) obtained from PCR+ individuals at the time of hospital admission. Statistical analyses were performed using two-way ANOVA. Asterisks represent $p$ values (****$p <$ 0.0001, *$p < 0.05$). Horizontal lines represent median values.

G18-145, BD Biosciences Cat#555787) and anti-human IgM-BV421 antibodies (1:1000) (Clone G20-127, BD Biosciences Cat#555783). The mix was incubated with the samples for 15 min at RT protected from light. One final wash was performed, and microbeads were resuspended in 200 μL of PBS supplemented with 5% FBS for acquisition. At least 600 events for each type of microbead were acquired in a FACSymphony flow cytometer (BD Biosciences) and geometric mean fluorescence intensities (gMFI) were obtained. Results were analysed using FlowJo version 10 (BD Biosciences).

**ELISA**. The protocol was adapted from a previously established immunoassay[13]. Briefly, 96-well ELISA plates (Nunc Maxisorp Cat#44-2404-21) were coated overnight at 4 °C with 50 μL of biotinylated RBD, S1, or N protein (Acrobiosystems) at 2 μg/mL (for RBD and S1) or 1.4 μg/mL (N) in PBS (Gibco). In some cases, recombinant S1 protein from human coronaviruses HCoV-NL63 or HCoV-229E (Sino Biological Inc., Cat#40600-V08H, 40601-V08H) were used to coat the plates. Then, the coating solution was removed and plates were blocked with 3% non-fat milk in PBST (PBS plus 0.1% Tween-20) for 1 h at RT. Serum samples were pre-diluted in 1% non-fat milk in PBST, and incubated for 2 h at RT. After three washes with 250 μL of PBST in a plate washer (Biotek), each well was incubated with an anti-human IgG-horseradish peroxidase (HRP) conjugated secondary antibody (1:5000) (GenScript Cat#A01854) or anti-human IgM-HRP (Novus Biologicals Cat#NBP1-75014) for 1 h at RT. Plates were washed three times with PBST, and 100 μL of TMB substrate (Thermo Scientific Cat#34021) was added to each well, incubated for 2 min, and the reaction was stopped with 50 μL of stop

solution (Thermo Scientific Cat#N600). The optical density (OD) was measured at 450 nm in a VictorNivo multimode plate reader (PerkinElmer).

**Neutralization assay**. Binding of RBD to recombinant ACE2 was measured by a commercial surrogate virus neutralization test (cPass™ SARS-CoV-2 Neutralization Antibody Detection Kit, Genscript)[33]. The percentage of inhibition was calculated with the following formula: $(1 − $ sample OD value/average preCOVID OD value$) \times 100$. Pearson correlation analyses between gMFI values obtained by C19BA and percentage of inhibition obtained by the neutralization assay for anti-RBD, anti-S1, and anti-N IgG antibodies were calculated using Prism 8 (GraphPad) considering a 95% confidence interval.

**Statistics and reproducibility**. To calculate the sample ODs and gMFIs, the values corresponding to the negative controls were subtracted from all samples. The AUC values were calculated as described in Amanat et al.[12]. Briefly, the background was set at 0.11 for each sample and AUC values were calculated using Prism 8 (GraphPad). The resulting values were divided by 100 and those that were below 1 were assigned a value of 0.5 for plotting purposes. Statistical analyses comparing different cohorts were performed with an unpaired two-tailed Student's $t$-test. Data were analysed using Prism 8 (GraphPad). Phylogram generated from the FASTA alignment file was performed using FastTree (https://www.genome.jp/). ROC curves and the corresponding AUC values were computed using the pROC R package (v.1.17.0.1)[34]. The values that maximized the Youden index were selected as the cut-off for reporting sensitivity and specificity values.

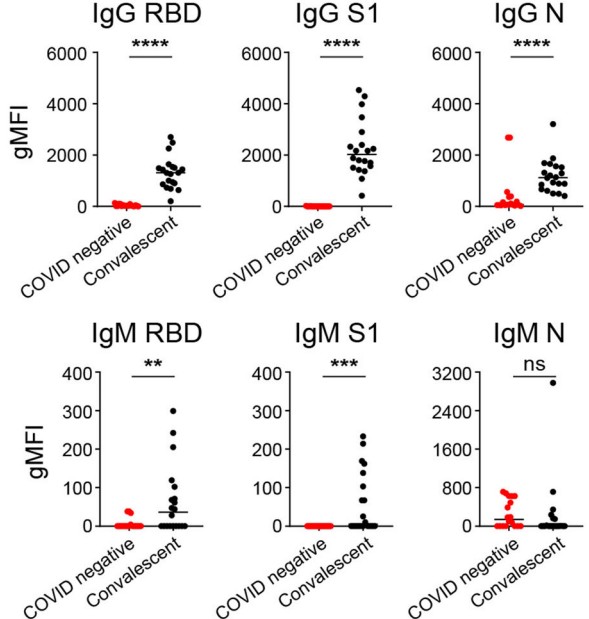

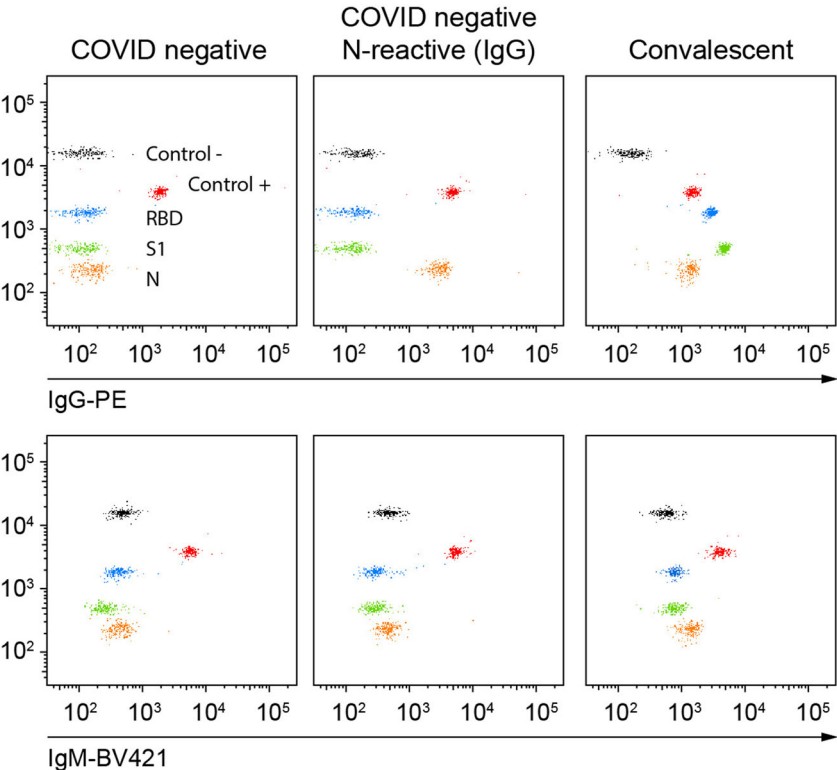

**Fig. 6 C19BA identifies IgG and IgM seroconversion in a cohort of samples from convalescent donors, and the presence of N-reactive IgG antibodies in Covid-seronegative samples. a** Serological responses against RBD, S1, and N measured by C19BA on serum samples ($n = 38$) from convalescent patients after at least 1 month of disease onset, IgG (top) and IgM (bottom) are shown. **b** Representative dot plots obtained by C19BA showing IgG (top) and IgM (bottom) levels, as indicated. Statistical analyses were performed using an unpaired two-tailed Student's $t$-test. Asterisks represent $p$ values (****$p <$ 0.0001, ***$p < 0.001$, **$p < 0.01$). Horizontal lines represent median values.

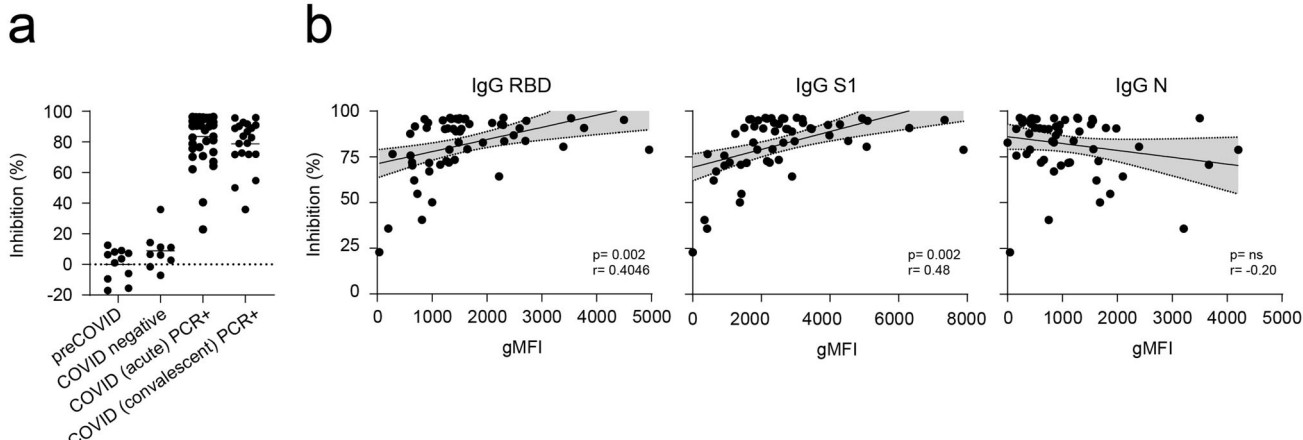

**Fig. 7 Levels of anti-RBD and anti-S1 IgG antibodies measured by C19BA correlate with neutralization capacity of serum samples. a** Neutralization assay performed on samples from different cohorts of individuals as indicated, including healthy and seropositive donors (*n* = 77). **b** Correlation of gMFI values obtained by C19BA and % Inhibition obtained by the neutralization assay for anti-RBD IgG (left), anti-S1 IgG (middle), and anti-N IgG (right) (*n* = 56). Pearson correlation coefficients (*r*) and their corresponding *p* values are shown.

**Reporting summary**. Further information on research design is available in the Nature Research Reporting Summary linked to this article.

## Data availability

All data generated or analysed during this study (and its supplementary information files) are included in this published article (Supplementary Data 1). Any remaining information can be obtained from the corresponding author upon reasonable request.

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

## Acknowledgements

We thank Petros Tyrakis and Iván Martínez-Forero for critical reading and editing of the manuscript. Support was provided by the Severo Ochoa Excellence Accreditation from MCIU (SEV-2016-0644) and the SPRI I+D COVID-19 fund (Gobierno Vasco). Personal fellowships: A.A.-V. (La Caixa Inphinit LCF/BQ/DR20/11790022), A.B. (AECC Bizkaia), A.G.d.R (Bikaintek), A.P. (Ramón y Cajal), B.J.-L. (Gob. Vasco), and E.P.-F. (Juan de la Cierva-Formación). M.L.M.-C. acknowledges RTC2019-007125-1, DTS20/00138, SAF2017-87301-R, and BBVA UMBRELLA project. M.L.-H. acknowledges the ISCIII for grant COV20-0170 and the Government of Cantabria for grant 2020UIC22-PUB-0019. O.M., J.-M.M., and N.G.A.A. acknowledge the Agencia Estatal de Investigación (Spain) for grants CTQ2015-68756-R, RTI2018-101269-B-I00, and RTI2018-095700-B-I00, respectively. A.P. has received grant funding from the European Research Council (ERC), grant agreement number 804236 (Horizon 2020), and the FERO Foundation.

## Author contributions

A.P. conceived and administered the project. L.E.-M. and A.B. performed all experiments and analysed data. A.B., L.E.-M, N.G.A.A R.G.-R., and E.P.-F. performed computational and statistical analyses. A.G.d.R., S.Y.L., B.J.-L., and A.A.-V. contributed ideas. O.M., J.M.M., C.B., M.B., N.E., M.L.M.-C., and M.L.-H. provided patient samples. A.P. wrote the manuscript with help from all co-authors.

## Competing interests

The authors declare no competing interests.
