## [Peer Review File · Communications Biology]

Reviewers' comments:

Reviewer #1 (Remarks to the Author):

The author developed an assay for detection of COVID 19 antibody using multiplex Luminex bead-based assay. Development of this type of the assay is needed to contain the COVID-19 infection during the pandemic time. However, author need to address the following issues to adapt this assay for serological testing.

Major Comments:

1. In the study the sample size is very small, and the author conducted the study with 43 symptomatic and PCR positive for SARS-2 viral RNA and 50 negative samples. Author should include more positive samples and mild/asymptomatic COVID-2 cases. The author must include the serum samples from other common corona, Influenza, respiratory HIV and CMV viral infection. If possible you can include some serum samples from TB bacterial infections. This will explain any cross reactivity if exists with respiratory diseases and COVID-19.
2. Sensitivity and specificity of this assay must be explained with ROC curves this will give very accurate sensitivity and specificity.

Minor Comments

3. Any co relation of the present assay developed by you and the COVID-2 neutralization assay
4. Several other studies observed the cross reactivity of N protein antibody of common corona and COVID-19 virus, the author mentioned about this cross reactivity in the discussion. Is there any justification to include the COVID-19 N protein in this multiplex assay?
5. Will you please explain how early you can detect the COVID-19 antibody after the infection using this assay?
6. Please explain the assay performance in single plex (using RDB, N and S1 protein separately) and in multiplex.
7. Whether your assay can be used as surrogate assay to determine the neutralizing antibody during plasma therapy without performing neutralization using live/VLP of SARS-2 virus.

Reviewer #3 (Remarks to the Author):

Egia-Mendikute et al. describe the development of a flow cytometry-based assay using beads to detect antibodies reactive to SARS-CoV-2 nucleoprotein, S1 domain of the spike or receptor binding domain of the spike. The bead-based assay shows the ability to detect antibodies (human IgG and IgM) to the three different antigens in a multiplex manner. In addition, the authors demonstrate improved sensitivity of assay to detect low levels of SARS-CoV-2 antigen specific antibodies in patient's sera compared to a gold-standard ELISA. Testing of samples taken before the pandemic started indicated that low levels of nucleoprotein specific antibodies are present in a small proportion of subjects. The flow-based assay is well described, however, some important points need to be addressed.

Major

1. Continuous line numbers should be added.
2. The authors state that the bead-based assay is more sensitive than ELISA. However, there are many different enzyme-linked immunosorbent assays available which vary in sensitivity and specificity. Please specify and add that the bead assay was superior to the one ELISA tested in this study.
3. N-reactive antibodies have been reported to decline faster after infection as compared to spike specific antibodies. "We reasoned that fully seroconverted individuals would present antibodies against the three chosen antigens." Depending on the timing and time passed after infection IgM antibodies as well as N-specific antibodies (both IgG and IgM) might be declining already and below the limit of

detection. Please discuss and mention details about the samples tested (e.g. (days after infection) in this study.

4. In the introduction the authors state that: "In general, ELISAs have an acceptable specificity and sensitivity profile for performing large epidemiological studies, but lack the required accuracy for routine SARS-CoV-2 serology^{21,22}. Moreover, ELISAs have important limitations including cost, time and throughput. They require optimization..."

This statement should be rephrased. Many large epidemiological studies have been done in recent weeks/months using ELISAs and many institutions are performing routine serology using ELISAs. How does the time and cost doing ELISAs compare to a flow-based assay that requires an expensive flow-cytometer as well as beads and preparation of beads? Moreover, optimization of an assay is part of developing and using any assay.

5. Figure 2 shows the result from one serum sample. It would be of interest to the reader to include more samples to see the performance of the bead assay in comparison to the ELISA.

6. Please add sample numbers tested and samples that turned out positive in the text. It seems like that 6 preCOVID samples (out of 50?) were positive for IgG N (figure 3B). It would be great to see more preCOVID samples tested to strengthen the finding of the study and get an idea of the percentage of people that have cross-reactive N-antibodies.

7. How were the AUC values calculated for the ELISAs in Figure 3B? The binding curves show no to very little reactivity, however the AUC values almost reach values of 1000 (e.g. for IgG N). Why did the authors decide to only show a very small proportion of the study samples in the graphs? Similarly, Figure 4 only shows 4 representative samples out of 100.

Minor

1. Many research laboratories or clinical laboratories at hospitals might not have access to flow cytometers or lack personal with flow experience. Using ELISAs or antibody test kits based on ELISA are easier to use and probably easier to implement.

2. How long after infection where the samples taken? 'At this early stage of infection, not all samples from the COVID cohort presented reactivity against viral antigens'. As mentioned above, please add the time passed after infection for those samples. Were the samples taken too early to detect seroconversion or is the assay not sensitive enough to detect antibodies, or did those individuals not mount much of an antibody response?

3. Sensitivity and specificity metrics to correctly classify seroconverters have not been established in this study. The authors can, however, comment on the sensitivity of the assays used in this study itself.

4. What effective protein concentration is used in the bead assay and how does that compare to the concentration (2ug/ml) used in the ELISA?

5. In figure 3 it looks like the dynamic range is better in the ELISA. (The curves are in the positive range and suddenly drop below the limit of detection in the bead assay)

6. In the ELISA biotinylated proteins were used. How was the biotinylation done and could biotinylation mask epitopes that reduce the sensitivity of the assay? Would the sensitivity change using non-biotinylated antigens?

7. Please indicate the cutoff value (e.g. dashed lines) in figure 2, 3 and supp figure 2. The authors state that for values below 0.11 the values were set to 0.11 in both the ELISA and bead-based assay. However, these two assays have different scales and readout (OD vs gMFI) and the cut-off should be specific to either assay.

Reviewer #1 (Remarks to the Author):

The author developed an assay for detection of COVID 19 antibody using multiplex Luminex bead-based assay. Development of this type of the assay is needed to contain the COVID-19 infection during the pandemic time. However, author need to address the following issues to adapt this assay for serological testing.

Major Comments

1.1) In the study the sample size is very small, and the author conducted the study with 43 symptomatic and PCR positive for SARS-2 viral RNA and 50 negative samples. Author should include more positive samples and mild/asymptomatic COVID-2 cases. The author must include the serum samples from other common corona, Influenza, respiratory HIV and CMV viral infection. If possible you can include some serum samples from TB bacterial infections. This will explain any cross reactivity if exists with respiratory diseases and COVID-19.

Response:

We thank the reviewer for the comments. In order to increase the n numbers in our study, we have included an additional 72 serum samples (54 COVID-19 + 18 negative controls) to the original 93 samples (43 COVID-19 + 50 pre-COVID). The new samples correspond to additional cohorts that include acute (mild/moderate and severe) and convalescent individuals, allowing for new analyses that are now included in the manuscript (new figures 5, 6 and 7).

We have also further explored potential specificity issues as a result of antibody cross-reactivity against the chosen SARS-CoV-2 antigens (RBD, S1 and N) in Covid-19 seronegative individuals. In order to address this point, we originally included a cohort of samples collected before the pandemic (2018), showing that RBD and S1 are specific for assessing seroconversion against SARS-CoV-2, while the nucleoprotein (N) can deliver false positive results. These findings are supported by the homology alignment of sequences of the Spike and N against other related coronaviruses, and in line with accumulating evidence in the literature supporting the superior specificity of RBD. (example: Premkumar, L. *et al.* The receptor binding domain of the viral spike protein is an immunodominant and highly specific target of antibodies in SARS-CoV-2 patients. *Sci Immunol* **5**, doi:10.1126/sciimmunol.abc8413 2020).

To further strengthen our specificity claims in the context of potential cross-reactivity of the chosen antigens with antibodies generated after other viral infections, we have now added data showing that serum samples of seropositive individuals against the spike of other coronaviruses do not cross-react with RBD or S1 in our assay (n= 80), (Supplementary Figure 3).

In addition, we have measured the specificity of our assay in a limited number (as many as we could obtain in the time window of this revision and supported by ethical approval) of serum samples corresponding to other pathologies, as suggested by the reviewer. These include: Salmonella sepsis, pancreatitis, renal insufficiency, and other respiratory tract illnesses. In these samples, we found no cross-reactivity against the Spike of SARS-CoV-2, a comparison against preCovid and convalescent Covid cohorts is shown here:

The individual values for each of these samples is shown here:

1.2) Sensitivity and specificity of this assay must be explained with ROC curves this will give very accurate sensitivity and specificity.

Response:

In order to calculate sensitivity and specificity values for this assay, we generated ROC curves including preCOVID samples (specificity) and Covid-19 samples confirmed by PCR (sensitivity), total n=84. These curves and values are presented in Supplementary figure 4.

Minor Comments

1.3) Any co-relation of the present assay developed by you and the COVID-2 neutralization assay

Response:

In order to address this suggestion, we have performed a commercial assay that measures the ability of serum samples to prevent the interaction of RBD and ACE2. To this end, we analysed 77 serum samples (including PreCOVID, Covid-negative, acute and convalescent cohorts) using the surrogate virus neutralization test (sVNT) cPass™ SARS-CoV-2 Neutralization Antibody detection Kit (GenScript):

We then performed correlation analyses between the values obtained on the neutralization assay (%Inhibition) and the levels of antibodies obtained by C19BA. Figure 7 now shows that the levels of

anti-RBD and anti-S1 IgG antibodies significantly correlate ($r=0.71$, 0.72 respectively; $p < 0.0001$) with the percentage of inhibition, while the levels of anti-N IgG antibodies presented a weaker correlation ($r=0.28$; $p = 0.0141$), as expected. These new results are presented on lines 151 to 161 and discussed on lines 188 to 190.

1.4) Several other studies observed the cross reactivity of N protein antibody of common corona and COVID-19 virus, the author mentioned about this cross reactivity in the discussion. Is there any justification to include the COVID-19 N protein in this multiplex assay?

We started this project on March 2020 and decided to leverage the multiplex capability of cytometric bead arrays to interrogate the antibody responses against this set of antigens and study and compare their sensitivity/specificity. At this early stage, several commercial serology assays based on ELISA or rapid LFA used N protein as antigen. In this context, we feel that reinforcing the potential of cross-reactivity associated with the N protein retains important value, especially when the majority of reports on the literature are focused on existing anti-N protein T cell responses in unexposed individuals, rather than antibody cross-reactivity (i.e. Le Bert N, Tan AT, Kunasegaran K, Tham CYL, Hafezi M, Chia A, Chng MHY, Lin M, Tan N, Linster M, Chia WN, Chen MI, Wang LF, Ooi EE, Kalimuddin S, Tambyah PA, Low JG, Tan YJ, Bertoletti A. SARS-CoV-2-specific T cell immunity in cases of COVID-19 and SARS, and uninfected controls. Nature. 2020 Aug;584(7821):457-462. doi: 10.1038/s41586-020-2550-z. Epub 2020 Jul 15. PMID: 32668444).

Moreover, this multiplexed assay is designed as a platform that could allow the interrogation of additional antigens on a modular manner. In this context, including certain epitopes of the N protein could have additional value/applications; for example, identifying vaccinated vs. naturally immunised individuals based on the pattern of responses against multiplexed antigens. As a note, our readout is not based in a closed system (i.e., Luminex), the readout can be performed in any flow cytometer (brand-agnostic) equipped with the required basic laser configuration.

1.5) Will you please explain how early you can detect the COVID-19 antibody after the infection using this assay?

Response:

We detect antibodies against RBD or S1 in serum samples obtained at time of hospital admission. This corresponds to 76 % of these acute patients (Supplementary Figure 2), which is in line with what is reported by longitudinal studies (Ren L, Zhang L, Chang D, Wang J, Hu Y, Chen H, Guo L, Wu C, Wang

C, Wang Y, Wang Y, Wang G, Yang S, Dela Cruz CS, Sharma L, Wang L, Zhang D, Wang J. The kinetics of humoral response and its relationship with the disease severity in COVID-19. *Commun Biol.* 2020 Dec 11;3(1):780. doi: 10.1038/s42003-020-01526-8. PMID: 33311543; PMCID: PMC7733479.). The percentage of patients detected as seropositive in an independent cohort of convalescent patients is 100% (Figure 6)

1.6) Please explain the assay performance in single plex (using RDB, N and S1 protein separately) and in multiplex.

Response:

The assay performance in singleplex is not expected to be different than in multiplex, with the consideration that in this combination of antigens (RBD, S1, N), RBD and S1 share the RBD sequence. This redundant design allows an intra-assay confirmation of binding and the identification of antibodies directed to non-RBD regions of the Spike. As a consequence of this redundancy, a theoretical decrease in the detection ability in the multiplex format compared to singleplex could be expected (i.e. antibodies against RBD will bind to two different beads, halving the amount of antibodies binding on each bead).

1.7) Whether your assay can be used as surrogate assay to determine the neutralizing antibody during plasma therapy without performing neutralization using live/VLP of SARS-2 virus.

Response:

This is an interesting application of the platform with important applications in the context of vaccination and emerging variants. We are planning the development of a neutralization assay based on this platform. In this approach, beads coated with the Spike (including variants) are incubated with serum samples to then perform the readout based on the binding ability of recombinant, fluorochrome tagged ACE2.

Reviewer #3 (Remarks to the Author):

Egia-Mendikute et al. describe the development of a flow cytometry-based assay using beads to detect antibodies reactive to SARS-CoV-2 nucleoprotein, S1 domain of the spike or receptor binding domain of the spike. The bead-based assay shows the ability to detect antibodies (human IgG and IgM) to the three different antigens in a multiplex manner. In addition, the authors demonstrate improved sensitivity of assay to detect low levels of SARS-CoV-2 antigen specific antibodies in patient's sera compared to a gold-standard ELISA. Testing of samples taken before the pandemic started indicated that low levels of nucleoprotein specific antibodies are present in a small proportion of subjects. The flow-based assay is well described; however, some important points need to be addressed.

Major Comments

(3.1) Continuous line numbers should be added.

Response:

We thank the reviewer for the encouraging comments. Continuous line numbers have been added to the manuscript.

(3.2) The authors state that the bead-based assay is more sensitive than ELISA. However, there are many different enzyme-linked immunosorbent assays available which vary in sensitivity and specificity. Please specify and add that the bead assay was superior to the one ELISA tested in this study.

Response:

We agree with the reviewer and our claims about the comparison to ELISA have been limited to the ELISA performed in our laboratory, avoiding generalization (lines 32, 98, 168 and 182).

(3.3) N-reactive antibodies have been reported to decline faster after infection as compared to spike specific antibodies. "We reasoned that fully seroconverted individuals would present antibodies against the three chosen antigens." Depending on the timing and time passed after infection IgM antibodies as well as N-specific antibodies (both IgG and IgM) might be declining already and below the limit of detection. Please discuss and mention details about the samples tested (e.g. (days after infection) in this study.

Response:

We have clarified the time passed between infection and the obtention of serum samples for acute (PCR+, samples obtained at hospital admission) and convalescent (one month after infection): manuscript lines 106, 132, 218-229, 425, 433, 441, 449)

(3.4) In the introduction the authors state that: "In general, ELISAs have an acceptable specificity and sensitivity profile for performing large epidemiological studies, but lack the required accuracy for routine SARS-CoV-2 serology^{21,22}. Moreover, ELISAs have important limitations including cost, time and throughput. They require optimization..."

This statement should be rephrased. Many large epidemiological studies have been done in recent weeks/months using ELISAs and many institutions are performing routine serology using ELISAs. How does the time and cost doing ELISAs compare to a flow-based assay that requires an expensive flow-cytometer as well as beads and preparation of beads? Moreover, optimization of an assay is part of developing and using any assay.

Response:

We have rephrased and deleted some of our previous claims (cost, assay development) about ELISA (line 59), highlighting the fact that in general only one antigen can be tested per well (lack of multiplex capability). We also now mention automated chemiluminescent immunoassays that are routinely used in the clinic, but also require specific equipment (line 41).

Regarding costs, we have decided not to include claims related to potential commercial aspects in order to maintain the scientific discussion unbiased. Flow cytometers have a high cost but are found in most hospitals, similar to plate readers. We also agree with the reviewer that, when commercial assays/kits are not available, assay development could be a time-consuming process.

(3.5) Figure 2 shows the result from one serum sample. It would be of interest to the reader to include more samples to see the performance of the bead assay in comparison to the ELISA.

Response:

Figure 2 represents the titration of different commercial antibodies against recombinant RBD, S1 and N-by flow cytometry and ELISA. In this revised version, we have significantly increased the number of samples: we have included an additional 72 serum samples (54 Covid-19 + 18 negative controls) to the original 93 samples (43 Covid-19 + 50 pre-Covid). The new samples correspond to additional cohorts

that include acute (mild/moderate and severe) and convalescent individuals, allowing for new analyses that are now included in the manuscript (new figures 5 and 6).

(3.6) Please add sample numbers tested and samples that turned out positive in the text. It seems like that 6 preCOVID samples (out of 50?) were positive for IgG N (figure 3B). It would be great to see more preCOVID samples tested to strengthen the finding of the study and get an idea of the percentage of people that have cross-reactive N-antibodies.

Response:

In order to address this comment, we have now included additional negative samples (n=18) to a total of n=68, and we now present the sample numbers (line 119 and line 148) and percentages related to N-reactivity (14 % of total in unexposed individuals, line 192).

(3.7) How were the AUC values calculated for the ELISAs in Figure 3B? The binding curves show no to very little reactivity, however the AUC values almost reach values of 1000 (e.g. for IgG N). Why did the authors decide to only show a very small proportion of the study samples in the graphs? Similarly, Figure 4 only shows 4 representative samples out of 100.

Response:

The original calculation of AUC had a minor error related to the calculation of the AUC corresponding to values close to zero. This issue has been fixed and the methodology of the calculation is now clearly presented in materials and methods (line: 281), and Figure 3B has been revised accordingly:

The AUC values were calculated as described in Amanat et al.¹². Briefly, the background was set at 0.11 for each sample and AUC values were calculated using Prism 8 (GraphPad). The resulting values were divided by 100 and those that were below 1 were assigned a value of 0.5 for plotting purposes.

We have also increased the number of representative samples shown (new Figure 5B, new Figure 6B).

Minor

(3.8) Many research laboratories or clinical laboratories at hospitals might not have access to flow cytometers or lack personal with flow experience. Using ELISAs or antibody test kits based on ELISA are easier to use and probably easier to implement.

Response:

We have deleted previous claims comparing this assay to ELISA in terms of implementation and costs, as suggested previously by the reviewer.

(3.9) How long after infection where the samples taken? 'At this early stage of infection, not all samples from the COVID cohort presented reactivity against viral antigens'. As mentioned above, please add the time passed after infection for those samples. Were the samples taken too early to detect seroconversion or is the assay not sensitive enough to detect antibodies, or did those individuals not mount much of an antibody response?

Response:

We have clarified the time passed between infection and the obtention of serum samples for acute (PCR+, samples obtained at hospital admission) and convalescent (one month after infection): manuscript lines 106, 132, 218-229, 425, 433, 441, 449).

For samples obtained from the acute COVID-19 cohort samples (obtained at time of hospital admission), we can detect antibodies against RBD or S1 in serum in 76 % of samples (Supplementary Figure 2). This is in line with what is reported by longitudinal studies (Ren L, Zhang L, Chang D, Wang J, Hu Y, Chen H, Guo L, Wu C, Wang C, Wang Y, Wang Y, Wang G, Yang S, Dela Cruz CS, Sharma L, Wang L, Zhang D, Wang J. The kinetics of humoral response and its relationship with the disease severity in COVID-19. *Commun Biol.* 2020 Dec 11;3(1):780. doi: 10.1038/s42003-020-01526-8. PMID: 33311543; PMCID: PMC7733479). This, together with the fact that we detect seroconversion in 100% of convalescent patients, suggest that negative acute samples were taken too early in order to detect seroconversion.

(3.10) Sensitivity and specificity metrics to correctly classify seroconverters have not been established in this study. The authors can, however, comment on the sensitivity of the assays used in this study itself.

Response:

In order to address this important comment, we carried out ROC curve analyses to calculate specificity and sensitivity values for our assay. These are presented in Supplementary Figure 4.

(3.11) What effective protein concentration is used in the bead assay and how does that compare to the concentration (2ug/ml) used in the ELISA?

Response:

A genuine advantage of the presented assay vs. ELISA is the required amount of recombinant protein per sample.

At the start of the project, we tested and optimised the amount of protein per bead. Our bead assay employs, per sample: (11.2 ng of S1, 4.1 ng of RBD and 5.1 ng of N) vs. ELISA (100 ng of RBD and S1, and 70 ng of N).

(3.12) In figure 3 it looks like the dynamic range is better in the ELISA. (The curves are in the positive range and suddenly drop below the limit of detection in the bead assay)

Response:

This is probably a visual effect resulting from comparing logarithmic vs linear graphs (causing this perceived drop), for example:

(3.13) In the ELISA biotinylated proteins were used. How was the biotinylation done and could biotinylation mask epitopes that reduce the sensitivity of the assay? Would the sensitivity change using non-biotinylated antigens?

Response:

Recombinant proteins used in this study leverage the Avitag technology, which consists in introducing a C-terminal tag (in the case of S1 and RBD), or N-terminal tag (Nucleoprotein). Avitag is a 15 aminoacids sequence that enables directed biotinylation, introducing a single terminal biotin molecule on each recombinant protein in a process mediated by the BirA enzyme. The terminal and additive nature of this biotin makes masking events unlikely, acting as a linker favouring the presentation of

the antigen on the surface of the beads. The ELISAs were performed with the same recombinant proteins that were used in the bead assay to allow a fair comparison between them. At the start of the project, we tried to coat the beads with non-biotinylated proteins (carboxy chemistry) but the coating efficiency varied between antigens (depending on size and amino acid composition). As a result, we used the biotin/streptavidin system, which does not present this issue.

(3.14) Please indicate the cutoff value (e.g. dashed lines) in figure 2, 3 and supp figure 2. The authors state that for values below 0.11 the values were set to 0.11 in both the ELISA and bead-based assay. However, these two assays have different scales and readout (OD vs gMFI) and the cut-off should be specific to either assay.

Response:

The methodology corresponding to the calculation of the AUC is now clearly described on line 281. We have also calculated the cutoff of each antigen for each technique, based on the ROC analyses presented on Supplementary Figure 4). The values are the following:

ROC cutoff FACS (gMFI):

- RBD-IgG: 30.75; S1-IgG: 7.9; N-IgG: 343.5
- RBD-IgM: 52.75 ; S1-IgM: 2.5 ; N-IgM: 0.5

ROC cutoff ELISA (OD):

- RBD-IgG: 0.051; S1-IgG: 0.109; N-IgG: 0.416
- RBD-IgM: 0.609; S1-IgM: 0.492; N-IgM: 0.406

We have decided not to include the dashed lines in the main Figures for the following reasons:

- For RBD and S1, the lines are not visible given their low value.
- For N, the lack of a genuine negative cohort limits the usefulness of including a cut-off for this antigen.

REVIEWERS' COMMENTS:

Reviewer #1 (Remarks to the Author):

Thanks for conducting the additional experiments and including the additional data to the original manuscript.

Reviewer #3 (Remarks to the Author):

The comments have been addressed appropriately and in detail, and new data has been added to the manuscript. A few minor points remain that need to be addressed.

- 1) Line 137: While significant differences could not be demonstrated in the ELISAs there is still the same trend visible as for the C19BA assay. Please rephrase.
- 2) For a meaningful correlation analysis (Figure 7) negative samples (e.g. negative in C19BA) should be excluded. In general, using % inhibition to correlate to gMFI seems to be not the best option as the % inhibition is restricted to values from 0-100. E.g. IC50 values might be more appropriate.
- 3) The commercial antibodies tested (Figure 2) are not referenced or described in the methods section. Please add more details.

REVIEWERS' COMMENTS:

Reviewer #1 (Remarks to the Author):

Thanks for conducting the additional experiments and including the additional data to the original manuscript.

Reviewer #3 (Remarks to the Author):

The comments have been addressed appropriately and in detail, and new data has been added to the manuscript. A few minor points remain that need to be addressed.

3.1) Line 137: While significant differences could not be demonstrated in the ELISAs there is still the same trend visible as for the C19BA assay. Please rephrase.

Response:

We agree with the reviewer, this statement has been replaced.

Line 137: The same trend can be observed by ELISA but did not reach statistical significance.

3.2) For a meaningful correlation analysis (Figure 7) negative samples (e.g. negative in C19BA) should be excluded. In general, using % inhibition to correlate to gMFI seems to be not the best option as the % inhibition is restricted to values from 0-100. E.g. IC50 values might be more appropriate.

Response:

We thank the reviewer for the comments. We agree with the suggestion of excluding Covid-negative samples from the correlation analyses. We originally included these samples to add a variety of cohorts in the neutralization assays. In order to present a fair correlation, these values have been excluded resulting in similar results: the presence of IgG antibodies against RBD and S1 significantly correlate with the neutralization capacity of the serum samples, as opposed to IgG antibodies against nucleoprotein. While we agree with the reasoning of IC50 being superior to %Inhibition for representing these results, we consider the later still valid for this purpose and well accepted in the literature. At this stage, calculating the IC50 for each sample would not be practical because of limiting availability of these particular samples (many finished in the previous revision), time and cost of neutralization assay kits.

3.3) The commercial antibodies tested (Figure 2) are not referenced or described in the methods section. Please add more details.

Response:

We thank the reviewer for pointing out this issue. Lines 246-248 describe the commercial antibodies used in Figure 2. "serially diluted commercial antibodies against RBD (GenScript Cat#A02038) or N (Acrobiosystems Cat#NUN-S41) starting from a 1 mg/mL concentration".